# Pathogenicity of La Jolla Virus in *Drosophila suzukii* following Oral Administration

**DOI:** 10.3390/v14102158

**Published:** 2022-09-30

**Authors:** Yvonne Linscheid, Tobias Kessel, Andreas Vilcinskas, Kwang-Zin Lee

**Affiliations:** 1Fraunhofer Institute for Molecular Biology and Applied Ecology, Branch of Bioresources, Ohlebergsweg 12, D-35392 Giessen, Germany; 2Institute for Insect Biotechnology, Justus Liebig University of Giessen, Heinrich-Buff-Ring 26-32, D-35392 Giessen, Germany

**Keywords:** *Drosophila suzukii*, La Jolla virus (LJV), *Iflavirus*, biological pest control

## Abstract

*Drosophila suzukii* (Ds) is an invasive pest insect that causes severe and widespread damage to soft fruit crops. Chemical control based on topical insecticides is inefficient and harmful to consumers and the environment, prompting interest in the development of biological control measures such as insect viruses with narrow host specificity. We previously described a strain of La Jolla virus (LJV) found in moribund Ds specimens in Germany. We demonstrated a pathogenic effect following the intrathoracic injection of LJV into adult Ds flies. However, the development of an effective biocontrol product based on LJV would require the characterization of (1) virulence following oral delivery, particularly in larvae, and (2) stability under different pH and temperature conditions reflecting realistic exposure scenarios. Here we describe the pathogenicity of LJV following oral delivery to Ds adults and larvae. The oral infection of Ds adults with LJV reduced survival in a concentration-dependent manner, whereas the oral infection of Ds larvae caused the arrest of development during pupation. LJV remained stable and infectious following exposure to a broad pH range and different temperatures. We, therefore, demonstrated that LJV is promising as a candidate biological control agent against Ds.

## 1. Introduction

Invasive alien species pose an increasing threat to biodiversity in Europe and also cause damage valued at €116.2 billion per year [1]. A prominent example is the spotted wing drosophila, *Drosophila suzukii* (Ds) Matsumura, which has become an invasive species on a global scale [2]. *D. suzukii* is endemic to Asia and was first described in Japan in the early twentieth century. In the 2000s, Ds initially spread to Hawaii and then to the west coast of the United States and Canada [3,4] as well as Europe [4]. Soft fruits, particularly berry crops such as blueberry and strawberry, are especially vulnerable to Ds, resulting in yield losses of 33–50% in the United States costing more than US$500 million per year [5]. Whereas most drosophilids prefer rotting fruit, Ds females are attracted to ripening fruit and possess a serrated ovipositor, which allows them to lay their eggs under the fruit skin [6,7]. The larvae hatch and feed within the fruit, where they are well protected. Females lay up to 400 eggs that can mature into flies within 8 days under ideal conditions, leading to population explosions [8,9]. The activity of Ds is highest between 15 and 20 °C [10].

The most common control method used against insect pests is chemical insecticides [11], but the wide use of broad-spectrum insecticides is damaging to human health and the environment [12]. Beneficial insects, including the natural enemies of pests, suffer the greatest impact [13]. Growers are therefore facing bans restricting the types of pesticides that can be applied. For example, Regulation (EU) 2021/1165 only allows a few pesticides to be used for fruit cultivation in Germany, most of which are oil-based fungicides or products derived from bacteria [14]. However, such products are generally less potent than chemical pesticides and do not affect larvae feeding inside ripe fruits. Innovative and sustainable methods are therefore needed to control pest insects without causing environmental damage. 

Crop yields can be maintained in the absence of chemical pesticides by deploying biological control methods based on insect pathogens, such as entomopathogenic bacteria and viruses [15]. For example, a biological control method has been developed against the codling moth (*Cydia pomonella*) based on *Cydia pomonella* granulovirus. This is particularly effective against larvae and prevents the infestation of apple trees [16]. Similarly, we have evaluated the ability of specific insect viruses to control Ds [17], which led to the isolation of La Jolla virus (LJV) from moribund Ds specimens in Germany [18]. LJV belongs to the family *Iflaviridae* and has a single-stranded positive-sense RNA genome 10,250 nt in length [19]. LJV was initially discovered in *Drosophila melanogaster* [20] before its isolation from Ds [18,21]. The intrathoracic injection of LJV caused significant mortality in adult Ds flies, suggesting its potential as a control method [18,19]. However, LJV must also be lethal when delivered orally, particularly to larvae, because this would be the typical administration route and target in the field following the application of a sprayable virus formulation.

Here we investigated the efficacy of orally delivered LJV, focusing on the stability of the virus under different pH and temperature conditions. The pH of the insect gut is species dependent and also varies according to the compartment. In Ds, an acidic central midgut is flanked by alkaline anterior and posterior regions, and it is therefore important to test LJV under these conditions to confirm it remains stable and infectious following ingestion. The thermostability of LJV is also an important property because the field performance of a sprayable formulation could be affected by the temperature. The determination of effective operational parameters for LJV will facilitate the development of formulations that can be used to control Ds in the field.

## 2. Materials and Methods

### 2.1. Fly Husbandry

The Ds strain used in this study was derived from a laboratory stock originally sourced from Ontario, Canada [22]. Flies were reared in a climate chamber at 26 °C with 60% humidity and a 12 h photoperiod. They were fed on a diet of 10.8% (*w/v*) soybean and cornmeal mix, 0.8% (*w/v*) agar, 8% (*w/v*) malt, 2.2% (*w/v*) molasses, 1% (*w/v*) nipagin and 0.625% (*w/v*) propionic acid.

### 2.2. Cultivation and Extraction of LJV 

LJV was initially isolated from moribund Ds flies and named LJV-Ds-OS20 (Carrau et al., 2021). We produced an LJV extract by injecting 1000 flies with ~46 nL of LJV suspension using a Nanoject II device (Drummond Scientific, Broomall, PA, USA), then incubating them in Ø 29 × 95 mm vials and fitting Ø 30 × 30 mm foam stoppers (Nerbe Plus, Winsen, Germany) with a fly diet for 3 days. We homogenized 250 flies in 100 mM HEPES (pH 8) for 5 × 2 min using 1.4 mm ceramic beads (MP Biomedicals, Eschwege, Germany) in a TissueLyser (Qiagen, Hilden, Germany). The homogenate was cleared by three cycles of centrifugation (12,000× *g*, 15 min, room temperature), transferring the supernatant to a fresh tube each time. The final supernatant was stored at −80 °C.

### 2.3. Oral Infection of Flies

Twenty female flies (3–7 days old) were fed LJV suspensions in 100 mM sucrose in 100 mM HEPES (pH 8) at concentrations of 10^6^ or 10^5^ genome equivalents (GE)/mL in a standard 2.5 cm vial containing a piece of folded paper towel. The negative control was 100 mM sucrose in 100 mM HEPES (pH 8) alone. The flies were maintained as described above and were provided with 100 µL 100 mM sucrose daily. Survivors were counted daily. The experiments featured three biological replicates, each consisting of three technical replicates. The results are presented as means of all nine replicates.

### 2.4. Oral Infection of Larvae 

Twenty second-instar (L2) Ds larvae were placed in 2.5 cm vials containing 2% agar and homogenized banana blended with 10^6^ GE/mL LJV in 250 µL 100 mM HEPES (pH 8). Negative controls were prepared by blending the homogenized banana with sterile buffer rather than the virus suspension. The survival and development of the larvae were monitored daily. We added 250 µL of the homogenized banana mix as food every 2 days. The experiments featured three biological replicates, each comprising three technological replicates. The results are presented as means of all nine replicates.

### 2.5. Standard Curve and Quantification of LJV 

A standard curve was prepared based on TaqMan qPCR experiments using serially diluted negative-sense RNA strands to measure the absolute concentration of LJV in the treated flies. We used a StepOne real-time PCR system (Applied Biosystems, Waltham, MA, USA) with the Luna Universal Probe One-Step RT-qPCR Kit (New England Biolabs, Ipswich, MA, USA) and the specific primer/probe set described in Table 1. Each reaction was heated to 55 °C for 10 min and 95 °C for 1 min followed by 40 cycles of 95 °C for 10 s and 60 °C for 30 s. To determine the LJV concentration, 50 ng/µL aliquots of RNA extracted from pools of five flies using TRIZOL reagent (Thermo Fisher Scientific, Waltham, MA, USA) were amplified by TaqMan qPCR. 

### 2.6. LJV Stability Test

For both the pH and thermostability test, the final concentration of LJV was adjusted to a high concentration level of 10^6^ GE/mL with 50 mM sucrose prior to feeding the flies. For negative and positive controls, the same settings as described in Section 2.3 were used. For the thermostability test, 400 µL of LJV in a 1.5 mL tube was incubated for 3 or 8 h at 20, 25, or 30 °C. For the pH stability test, the virus suspension was diluted 10-fold in 0.1 M sodium bicarbonate citrate buffer titrated to pH 1, 3, or 5, or in 0.1 M sodium bicarbonate buffer titrated to pH 7.5, 9, 10.5 or 12, and incubated at 21 °C for 15 min. The treated virus suspensions were then diluted 10-fold in 50 mM sucrose to neutralize the various pH conditions and then used for oral infection experiments carried out as described in Section 2.3, with the same number of replicates and controls. In the pH stability test, a separate cohort of flies was also injected with 46 nL of buffer at different pH values to exclude any negative effects caused directly by the pH of the buffer. 

### 2.7. Statistical Analysis 

Data were analyzed using GraphPad Prism v9.1.2 for Windows (GraphPad Software, San Diego, CA, USA). Survival curves were plotted using log-rank analysis (Kaplan–Meier method). The data from LJV-fed larvae were analyzed by two-way analysis of variance (ANOVA) with significance defined as follows: * *p* < 0.05, ** *p* < 0.01, *** *p* < 0.001, **** *p* < 0.0001.

## 3. Results

### 3.1. Standard Curve for the Quantification of LJV Based on TaqMan qPCR

Absolute quantification of the viral load following the oral administration of LJV to flies was achieved by preparing a standard curve using a highly sensitive TaqMan probe and specific primers (Table 1). The standard curve was generated by a fivefold serial dilution (Figure 1). The lowest dilution detected by the standard curve corresponded to 4 GE/mL and the highest dilution corresponded to 4 × 10^6^ GE/ mL. The PCR efficiency (R^2^) of the standard curve was 98% with a slope value of 2. The high R^2^ value indicated a broad dynamic range of detection, confirming that the standard curve achieved an accurate calculation of the absolute number of LJV RNA copies.

### 3.2. The Oral Administration of LJV Has Adverse Effects on Adult Flies and Larvae

Survival analysis following the oral administration of LJV suspensions to female flies revealed clear concentration-dependent effects (Figure 2). The high concentration (HC) of 10^6^ GE/mL resulted in a mean survival time of 5 days, whereas the low concentration (LC) of 10^5^ GE/mL resulted in a longer mean survival time of 10 days. Controls fed on the sterile buffer survived for a mean time of 15 days. Kaplan–Meier analysis revealed highly significant differences in survival rates between treated flies and controls (**** *p* < 0.0001). 

The absolute quantification of LJV showed that the virus titer increased substantially following oral administration, and was again dependent on the initial concentration. The HC of 10^6^ GE/mL resulted in an increase in three log values over 3 days (Figure 3a), whereas the LC of 10^5^ GE/mL resulted in an increase in two log values (Figure 3b).

The oral administration of LJV to DS larvae did not affect larval survival but had a profound effect on the frequency of pupation. In the control group fed on a mixture of HEPES and sucrose, all larvae pupated and 80% hatched as adult flies within 2 weeks (Figure 4, black and gray lines). This non-infected state consisted of three peaks, the first representing the larvae, the second representing the pupae, and the third (reaching a plateau) representing the adults. In the group fed on LJV, all larvae pupated but only 35% developed into adult flies, indicating developmental arrest and/or an increase in virulence. The difference between LJV-fed larvae and controls in terms of adult emergence was significant at *** *p* < 0.001.

### 3.3. Effect of pH on LJV Pathogenicity

We tested the stability of the virus within the pH range 1–9 by incubating it in different buffers and then using the treated suspensions for oral infection assays (Figure 5). As described above, we also tested a negative control group fed on a sterile mixture of HEPES and sucrose and a positive control group fed on the untreated LJV extract. Flies in the negative control group survived for an average of 17 days, whereas those in the positive control group survived for an average of 5 days. The flies in the experimental groups fed on virus suspensions treated at pH 1, 3, 5, 7.5, and 9 survived for 7–11 days. Those fed on virus suspensions treated at pH 10.5 and 12 survived for 15–17 days, similar to the negative control, suggesting that extreme alkaline treatment inactivates the virus. To ensure that the negative effects were caused by the virus and not by the extreme pH of the solution, we also injected another group of flies with buffer alone at each pH value. There was no statistically significant difference between these buffer controls and the HEPES/sucrose negative control, confirming that the observed effects were solely caused by the virus.

Quantification of the virus load revealed a strong increase in titers (2–5 log values) during the first 2–3 days of infection, although the titers were comparatively low when the virus was pre-treated at pH 1. The titer continued to increase on day 3 for flies fed on the virus pretreated at pH 3, 10.5, or 12, whereas the titer peaked on day 2 for flies fed on the virus pre-treated at pH 5, 7.5, or 9, as well as the positive control (Figure 6).

### 3.4. Effect of Temperature on LJV Pathogenicity

We tested the thermostability of LJV by incubating it at 20, 25, or 30 °C for 3 or 8 h and then using the treated suspensions for oral infection assays, along with the same negative and positive controls used in the other assays described above. Flies in the negative control group (without LJV) survived for an average of 16 days, whereas those in the positive control group (LJV with no pre-treatment) survived for an average of 5 days. Flies fed with LJV incubated at 20 °C for 3 and 8 h survived for an average of 6 and 7 days, respectively. Flies fed with LJV incubated at 25 °C for 3 and 8 h survived for an average of 8 and 10 days, respectively. Flies fed with LJV incubated at 30 °C died after 8 days regardless of the duration of incubation (Figure 7).

Absolute quantification revealed a strong increase in titers (2–4 log values) during the first 2–3 days of infection. The titer continued to increase on day 3 for most of the experimental groups, but there was a slight drop in the group exposed to the virus following pre-treatment at 20 °C for 3 h, as well as the positive control (Figure 8).

## 4. Discussions

We set out to determine the pathogenic effects of LJV following oral delivery to Ds adults and larvae, focusing on the effect of pH and temperature. The oral delivery of LJV significantly reduced the lifespan of Ds compared to control flies presented with a sterile medium. No other signs or symptoms of LJV-treated flies were detected. We found that all flies exposed to LJV succumbed earlier than control flies, and the effect was concentration-dependent. The virulence of LJV correlated strongly with the initial infective dose. The one log difference between oral doses of HC (10^6^ GE/mL) and LC (10^5^ GE/mL) LJV resulted in a doubling of the survival period, reflecting differences in the initial viral load and the titers measured by TaqMan qPCR during the course of the infection. We previously reported similar results when injecting LJV into the thorax of flies [18]. The development of a biocontrol agent based on LJV requires a field formulation with a virus concentration sufficient to kill Ds rapidly. Higher initial concentrations should be tested in future studies once a suitable in vitro system has been established for the mass production of virus particles. 

Having demonstrated the ability of LJV to kill adult Ds, we investigated the effect against larvae, which would be the major target under field conditions. Even if the eggs are oviposited under the fruit skin by Ds females, the larvae are in contact with the fruit surface from time to time, since the larvae are dependent on oxygen uptake for respiration. The gas exchange occurs through their posterior spiracles, the terminal openings from the tracheae, representing their respiratory system. This is supported by the observation during pupation, where larvae pupate with contact with the outer environment and their spiracles or respiratory valves are reaching out of the fruit. In addition, the contact of the Ds flies on the treated fruits would have an insecticidal effect. The oral delivery of LJV to larvae achieved similar, albeit attenuated effects compared to the adult stages. All LJV-fed larvae were pupated, but only 35% emerged as adults, compared to 80% of the untreated controls. The limited effect against larvae is likely to reflect the morphological and physiological differences between these developmental stages. A morphological structure conferring resistance to viral infection through the oral route is the peritrophic matrix (PM), a semi-permeable structure consisting of chitin microfibrils, proteins, and proteoglycans, forming a sheet between the intestinal epithelium and gut lumen. The PM serves as a physical barrier against pathogens and abrasive food components [23]. It has been reported that adult and larval *D. melanogaster* deficient for the PM structural protein Cristallin were more susceptible to oral DCV infection [24]. It has been suggested that changes in PM structure over time could reduce the barrier function against pathogens [25] resulting in higher susceptibility against ingested pathogens. For example, in adult flies the permeability of the gastrointestinal wall increases with age, making it easier for viral particles to cross the intestinal epithelium and then spread through the body [24]. Structural differences in the PM composition between larval and adult Ds might be one factor responsible for slight attenuated LJV susceptibility in larvae and should be investigated in future studies. Even so, the similar effect of LJV against different life stages of Ds will facilitate the development of biocontrol agents.

The previous investigation of viruses as oral control agents for drosophilid pests has achieved only limited success. *D. melanogaster* was orally exposed to Drosophila C virus (DCV), a single-stranded positive-sense RNA virus belonging to the family *Dicistroviridae* [24,26,27], and to Nora virus, an unclassified picorna-like virus that is persistently transmitted via the fecal-oral route [28]. In contrast to our results with LJV, DCV killed only 25% of flies in 20 days [27] and required a much higher initial concentration to do so—a 50% tissue culture infectious dose (TCID_50_/mL) of 10^11^—as well as the use of all pathway mutants with weaker immunity. Nora virus (found in *D. melanogaster* laboratory stocks) showed no detrimental effects on its host following oral infection (Habayeb, 2009). Similarly, bumblebees orally infected with three dicistroviruses (Israeli acute paralysis virus, acute bee paralysis virus and Kashmir bee virus) suffered only mild reductions in fitness despite the viral titer being two orders of magnitude higher than in our study [29]. When the authors tested doses similar to those used in our experiments (10^6^ GE), they were unable to detect active infection. However, injecting the same viruses directly into the thorax killed the bumblebees rapidly [29]. LJV, therefore, appears to be the most potent and effective virus against drosophilid species, causing a significant reduction in survival following the oral administration of doses as low as 10^6^ GE.

The stability of viral particles is a key requirement for the development of viruses as biocontrol agents. We tested the stability of LJV under different pH conditions reflecting the intestinal milieu of Ds, and different temperatures representing the climate range likely to be encountered in the field. The virus was incubated in buffers spanning a wide pH range and at three different temperatures for 3 or 8 h, before testing in oral feeding assays as above. LJV remained infectious in highly acidic and alkaline buffers, although the time required to establish an infection doubled following treatment in acidic buffers (pH 1–5) and tripled following treatment with alkaline buffers (pH 10.5). The most extreme alkaline treatment (pH 12) appeared to cause complete virus inactivation. Previous studies have shown that members of the family *Iflaviridae* release their genomic RNA more rapidly under acid conditions because the pH affects their capsid structure [30,31]. This agrees with our results showing that flies succumbed faster in the acidic range (pH 1–5) compared to alkaline conditions even though the viral load was lowest at pH 1. Previous studies with Triatoma virus belonging to the insect virus family Dicistroviridae [32] indicate that pH 8–9 is sufficient to destabilize the capsid subunits of the virus and release the viral genome. The discrepancy between detected viral genomes and killing efficiency could be due to the breakdown of viral capsid under strongly acidic or alkaline conditions, releasing viral genome. Hence, a high viral proliferation could just indicate a free viral genome free to be detected by our qPCR methodology with decreasing the number of infective viral particles. Finally, the analysis of temperature-dependent pathogenicity showed that incubation at different temperature regimes had only a minor effect compared to flies fed with untreated LJV, suggesting relatively stable viral particles in the tested temperature ranges. 

To conclude, we have shown for the first time a significant reduction in the lifespan of Ds following the oral administration of LJV, in contrast with the minimal impact of oral infection with DCV or Nora virus in other drosophilids such as *D. melanogaster* [24,26,27,28]. Furthermore, the oral administration of LJV to Ds larvae arrested metamorphosis, resulting in extensive death during pupation. The insecticidal effect of LJV in Ds larvae and adults provides dual use possibilities. Interestingly, a higher initial concentration of LJV accelerated its effects, killing almost all infected insects within ~5 days. The testing of LJV under different pH and temperature conditions revealed that the virus is extremely stable, which is a prerequisite for the development of formulations suitable for the scalable field application of sprayable biopesticides [18]. This will help to reduce the economic damage caused by Ds [33,34]. Most iflaviruses cause asymptotic infections in insects, with only a few studies indicating a detrimental effect on silkworms and honeybees [35,36]. We have demonstrated that LJV is an exceptional virus that is atypically effective against Ds, making it an ideal candidate for the development of a sustainable and environmentally friendly insecticide against this aggressive pest species.

## Figures and Tables

**Figure 1 viruses-14-02158-f001:**
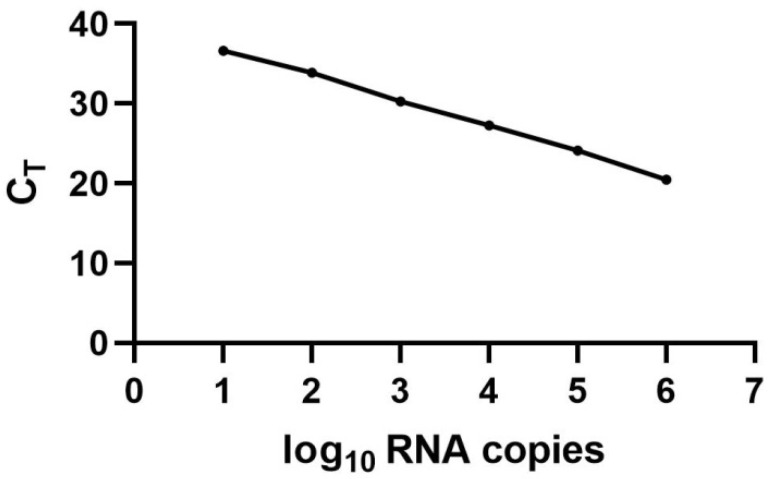
Standard curve of LJV. The *y*-axis shows the cycle threshold value (C_T_), which represents the number of PCR cycles needed to reach the threshold of the fluorescent signal. The *x*-axis represents the log_10_ of the absolute number of LJV RNA copies. The PCR efficiency is 98%, and the limit of quantification is 4 copies of the template corresponding to a C_T_ value of 37.

**Figure 2 viruses-14-02158-f002:**
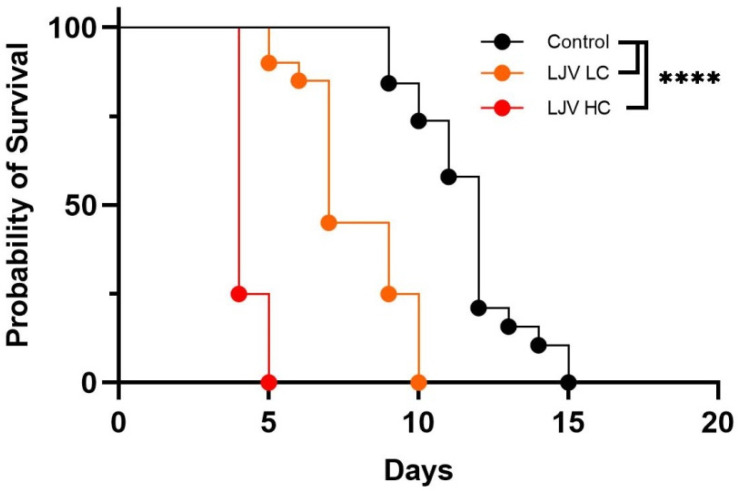
Survival of female Ds flies following the oral administration of LJV. Black line = untreated control, orange line = LJV low concentration (LC) of 10^5^ GE/mL, and red line = LJV high concentration (HC) of 10^6^ GE/mL. There was a significant difference in survival between LJV-fed flies and the untreated control (**** *p* < 0.0001).

**Figure 3 viruses-14-02158-f003:**
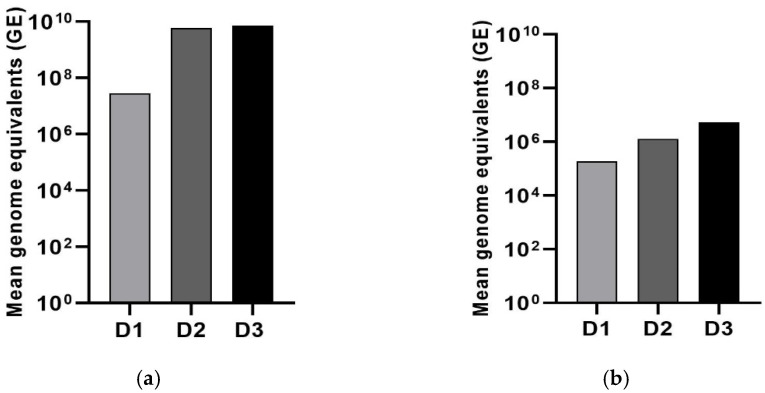
The quantification of LJV following oral administration to adult Ds flies. (**a**) High concentration (LC, 10^6^GE/mL). (**b**) Low concentration (HC, 10^5^ GE/mL). T-test analysis revealed no significant differences between the virus load measured on day one compared to day two or three.

**Figure 4 viruses-14-02158-f004:**
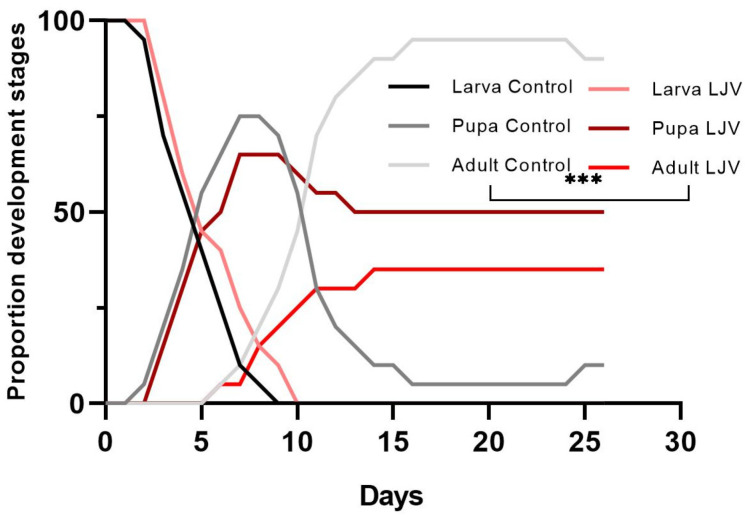
Characteristics of Ds larvae following the oral administration of LJV. The untreated control group is shown in shades of gray (black = larvae, dark gray = pupae, and light gray = adults). The LJV-infected group is shown in shades of red (pink = larvae, dark red = pupae, bright red = adults). The data were analyzed by two-way ANOVA. Statistically significant differences are shown at *** *p* < 0.001.

**Figure 5 viruses-14-02158-f005:**
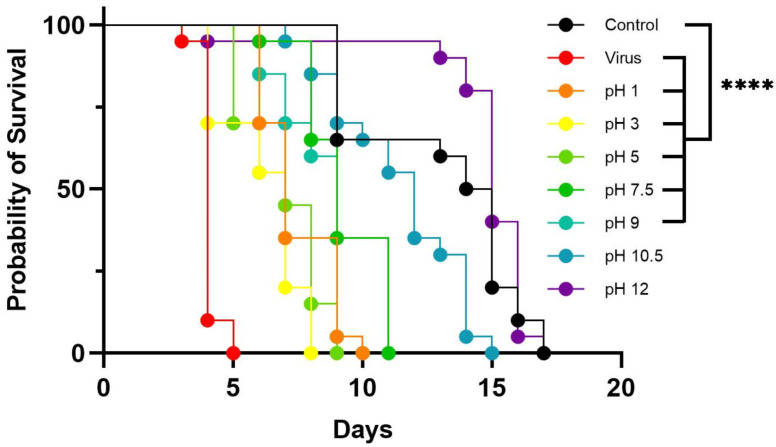
Survival of flies after exposure to LJV pre-treated in buffers over a broad pH range. Black = untreated negative control, red = LJV-fed positive control without pretreatment. LJV high concentration (HC, 10^6^ GE/mL) was used. Other lines represent LJV-fed flies following different pH treatments, as shown in the color key. Statistical analysis (Kaplan–Meier) indicated a significant difference (**** *p* < 0.0001) between the negative control and flies exposed to LJV following all pre-treatments except pH 10 and 12.5.

**Figure 6 viruses-14-02158-f006:**
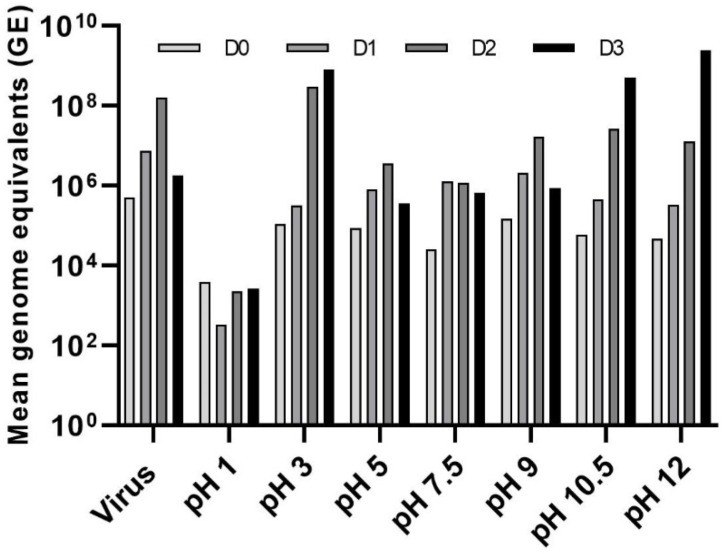
Absolute quantification of LJV following pH pre-treatment and oral delivery to adult Ds flies. The control (LJV without pH pre-treatment) is shown on the x-axis as “Virus” followed by the pH pre-treatment groups. The viral load was measured from day 0 to day 3 (D0–D3).

**Figure 7 viruses-14-02158-f007:**
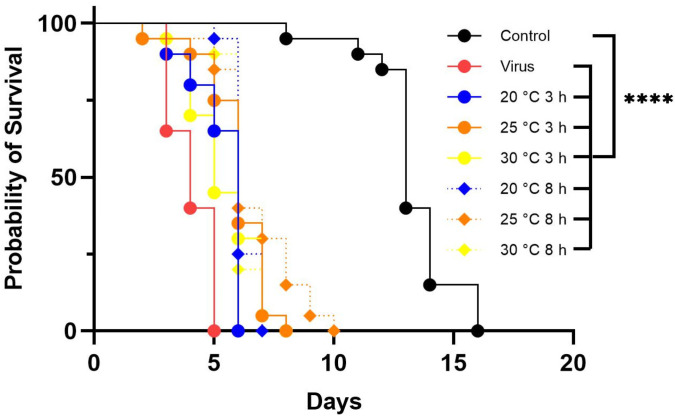
Survival assay following the oral delivery of LJV pre-treated at different temperatures. LJV high concentration (HC, 10^6^ GE/mL) was used. LJV was incubated at 20, 25, or 30 °C for 3 h (solid lines) or 8 h (dashed lines). Statistical analysis (Kaplan–Meier) indicated a significant difference (**** *p* < 0.0001) between the negative control and all flies exposed to LJV.

**Figure 8 viruses-14-02158-f008:**
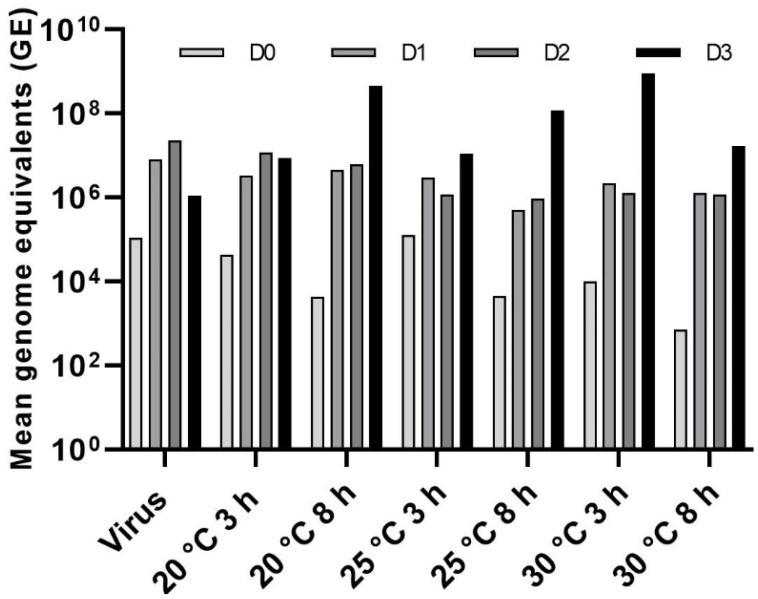
Absolute quantification of LJV following temperature pre-treatment and oral delivery to adult Ds flies. The control (LJV without pre-treatment) is shown on the x-axis as “Virus” followed by the temperature pre-treatment groups. The viral load was measured from day 0 to day 3 (D0–D3).

**Table 1 viruses-14-02158-t001:** Probes and primers used in this study.

Description	Sequence	Product Length (bp)
LJ specific probe *	5′-ACTCGGCGTTATCGTTACAACCGCACATATC-3′	
LJV forward primer	5′-CAACACGTTGTGCTGCCTGA-3′	128
LJV reverse primer	5′-TCCATCCAAACTCCACCTCC-3′	128

* labeled on 5′ with fluorescent reporter dye FAM, on 3′ with fluorescent quencher TAMRA. The probe is at position 64–95 bp within the 128 bp product.

## Data Availability

The data presented in this study are available on request from the corresponding author.

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
