# Peer review of "Pathogenicity of La Jolla Virus in Drosophila suzukii following Oral Administration"

_viruses, 2022, doi:10.3390/v14102158_

Round 1
Reviewer 1 Report
The manuscript by Linscheid et al. describes work to evaluate the suitability of La Jolla virus as a biological control agent against Drosophila suzukii. The manuscript is very well written, presented, and analyzed. This type of study is sorely needed and typically lacking. I commend the authors for their efforts. The manuscript was a pleasure to read. I had one observation that perhaps the authors could address. At lines 62 and 252, the authors state that larvae would be the target of the virus. However, earlier in the introduction the authors indicated that eggs are laid beneath the surface of fruit. Wouldn’t that make an orally delivered formulation useless against this pest? The authors do observe mortality in adult flies. I suppose the virus would not act sufficiently fast against adults to prevent egg laying. Is this correct? Perhaps the authors could address this inconsistency in the discussion.
Reviewer 2 Report
This manuscript presents the results of oral impact of Drosophila suzukii against ifravirus (single stranded RNA virus), and stability of those viruses under different pH and temperature conditions.  D. suzukii is economically important pest in US and Europe, and development of alternative control means other than chemical pesticides are needed. Using La Jolla virus (ifravirus) is one of the option as biological control of this species and the objectives of this study is reasonable. However, this manuscript cannot be published because of insufficient explanation of the data. It is expected to improve this manuscript by answering the following questions and enhance explanation of the data to be published in this journal.
Here are major comments.
1) For effect of pH and temperature experiments, the virus load was studied in Fig 6 and Fig8, respectively. Howerver, they were not discussed enough. There are some mysteries that the virus treated with pH12, for example, did not kill the hosts, but the virus loads were increased as much as (or more than) non-treated “virus” which kills the host for 5 days. This implies that the proliferation of the viral load within the host is not related to its host lethality. In this paper, the virus propagation was only done by qPCR. Other information, such as signs and symptoms of inoculated insects, should also be examined and described.
2) Virus propagation peaks were observed in Fig 6 on Day 2 but not in Fig 3. What is the difference in Fig 6 and 3?
Here are minor comments.
1) 2.5cm tube (L83) or vials (L92) were used. Are they the same things? What are they? It should be mentioned more about the container (volume, height, cap, etc.)
2) L82 Injection was done by what kind of device? It should be mentioned concretely.
3) L95 Biological replicates and technological replicates should be mentioned clearly.
3) 2.6 LJV stability
The positive control “virus” was not well written. As mentioned above, the “virus” in Fig. 5 is the same with HC in Fig. 2? So, how was pH in the virus inoculum used in Fig 2?
4) pH treatment was done for only 15 min. But there must be some time interval between pH treatment and starting the bioassay. After treatment with different pH buffer, are they neutralized? Otherwise, how could you say that the treatment were done for 15 min precisely?
5) The mean was used in Kaplan Meier assay data but why not used median?
Reviewer 3 Report
In this study, the authors showed that La Jolla Virus (LJV) is effective against Drosophila suzukii . The authors hypotheized that LJV is an ideal candidate for the development of an acceptable insecticide against this aggressive pest species
Major comments
Table 1: footnote: please correct Fam to FAM, and Tamra to TAMRA.
Also please add the position of probe inside 128 bp product
Figure 1: Standard curve:
a) please add the sensitivity and specificity for this assay
b) limit of detection and limit of quantification.
Figure 2: It is not convincing that the authors used 10^6 Genome as high conc and 10^5 Genome as a low conc. Why the authors did not use 10^3 or 10^2 Genome as a low conc?. How did the authors decide the high and low conc used?
Figure 3: It is surprisingly that low conc showed high efficiency replication than high conc as shown by mean genome equivalent D1- D3. Please verify.
Please go back to my previous comments. Why the authors did use low conc 10^2 or 10^3 as an infectious dose?
Please add statistic in figure 3.
Figure 5: what infectious dose the authors used? low or high. Please mentioned the details
Figure 6: the results are not explained well and not matched with figure 5. It is not clear why higher viral load detected at PH 3, 10.5, and 12 , while there is a significant difference in the survival curve especially at PH 3.
Figure 7:what infectious dose the authors used? low or high. Please mentioned the details
Round 2
Reviewer 3 Report
No further comments